# Effects and predictors of intravenous lidocaine infusion for patients with fibromyalgia

**Min Liu[1], Stephany Harris[1], Anna P. Andreou[1,2], Adnan Al-Kaisy[1], David Pang[1], Xuenong Bo[3]***

**1** Pain Management and Neuromodulation Centre, Guy's and St. Thomas' Hospital, London, United Kingdom, **2** Headache Research, Wolfson Sensory, Pain and Regeneration Centre, Institute of Psychology, Psychiatry and Neuroscience, King's College London, London, United Kingdom, **3** Centre for Neuroscience, Surgery and Trauma, Queen Mary University of London, London, United Kingdom

* x.bo@qmul.ac.uk

## Abstract

### Objective

Fibromyalgia is a chronic pain condition characterised by widespread pain. The current treatment primarily focuses on self-management and symptomatic relief. IV lidocaine infusion is the most performed procedure in the UK that is offered after conventional therapy has failed. We aim to identify the predictors of response to systemic lidocaine to enable targeted treatment for individuals who are more likely to benefit.

### Methods

This study adheres to the RECORD guidelines for reporting studies using routinely collected observational data. It was conducted retrospectively and employed data derived from clinical records of patients who received standard care. Adult patients who had completed questionnaires and quantitative sensory testing (QST) before IV lidocaine infusion were included. We collected data consecutively from 132 patients, including 24 men and 108 women. Responders were defined as patients who experienced a pain reduction of 50% or greater lasting for at least three weeks following an IV lidocaine infusion at a dose of 5 mg/kg.

### Results

We identified 22% of patients as responders. Our findings indicate a notable gender disparity in the number of responders, with a response rate of 25.9% observed in female patients compared to 4.2% in male patients ($p = 0.02$). Functional impairment in the revised fibromyalgia impact questionnaire was higher in female patients than male patients ($22.6 \pm 5.8$ vs $19.8 \pm 7.7$, $p = 0.043$). Responders were younger ($42.7 \pm 11.2$ vs $49.4 \pm 11.4$, $p = 0.003$), had shorter pain duration in years ($10.0 \pm 6.1$ vs $14.1 \pm 9.3$, $p = 0.015$), and lower weekly pain scores ($7.8 \pm 1.7$ vs $8.5 \pm 1.4$, $p = 0.014$).

**Data availability statement:** All relevant data are within the manuscript and its Supporting information files.

**Funding:** The author(s) received no specific funding for this work.

**Competing interests:** The authors have declared that no competing interests exist.

**Abbreviations:** ADP, average daily pain; AWP, average weekly pain; CDT, cold detection threshold; CDP, cold pain threshold; DMA, dynamic mechanical allodynia; DSIS, daily sleep interference score; HPT, heat pain threshold; LDP, least daily pain; MDT, mechanical detection threshold; MPS, mechanical pain sensitivity; MPT, mechanical pain threshold; PGIC, patient global impression of change; QST, quantitative sensory testing; rFIQ, revised fibromyalgia impact questionnaire; SSS, symptom severity score; TSL, thermal sensory limen; WDP, worst daily pain; WDT, warm detection threshold; WPI, widespread pain index; WSIS, weekly sleep interference score; WUR, wind-up ration.

No significant difference in QST parameters or loss/gain phenotypes was observed between responders and non-responders. A normal QST was identified in 25% of responders and 16% of non-responders. Consequently, QST alone could not predict the response to systemic lidocaine infusion.

## Conclusions

IV lidocaine infusion proves effective, especially for younger female patients, which should be added to conventional therapies for these patients.

## Introduction

Fibromyalgia is a chronic condition characterised by widespread musculoskeletal pain that affects 2–8% of the world population [1]. It is associated with fatigue, cognitive function impairment, mood and sleep disturbance. Evidence has revealed complex mechanisms including altered sensory and pain processing in the central nervous system [2], changes in the peripheral nervous system such as small fibre pathology [3–5], dysregulation of immune processes [6] and the presence of autoantibodies [7]. Commonly used medications, including gabapentinoids, tricyclic antidepressants, serotonin and noradrenaline reuptake inhibitors such as duloxetine, have modest effects on fibromyalgia pain [8]. Lidocaine, an amide local anaesthetic and anti-arrhythmic agent, has proved to be a valuable alternative for refractory chronic pain syndromes [9]. Evidence has shown the effectiveness of systemic lidocaine in treating fibromyalgia. An earlier randomised pilot study has found that IV lidocaine provides additional benefits for patients receiving conventional medications [10]. The analgesic effect of lidocaine occurs during and after the infusion, which could last for 3 weeks or longer [11–15].

The IV lidocaine infusion was the most performed procedure for fibromyalgia patients, accounting for approximately 6,000 cases each year in England [16]. Evidence has shown that lidocaine exerts a modality-specific effect rather than a general pain-relieving effect. Lidocaine significantly reduced spontaneous pain, the intensity of brush-induced allodynia and mechanical hyperalgesia, but was no better than the placebo against thermal allodynia and hyperalgesia in patients with central neuropathic pain after stroke [17]. Lidocaine reduced spontaneous pain and significantly reduced response to stroking and cold stimuli in patients with complex regional pain syndrome (CRPS) [18]. Identifying predictive factors allows us to give treatment to those likely to respond. This study aims to improve systemic lidocaine effectiveness and reduce costs for non-responders.

Quantitative sensory testing (QST) can provide information regarding large myelinated Aβ, thinly myelinated Aδ, and small unmyelinated C fibre functions, and their corresponding central pathways. It is a non-invasive method to assess the loss and gain of sensory function, which can enhance our understanding of pathophysiological mechanisms [19]. QST measures have consistently differentiated fibromyalgia from localised chronic lower back (CLB) or joint pain. Fibromyalgia patients show

increased sensitivity to thermal and mechanical pain across the body, unlike the localised pressure pain sensitivity in CLB participants and the near-normal sensory profile in osteoarthritis. Sensory abnormalities indicate changes in somatosensory processing and pain mechanisms. QST also identifies subtypes and predicts treatment response. Two fibromyalgia subgroups were identified based on cold and heat pain thresholds [22] and four phenotypes were suggested using a combination of QST and corneal nerve fibre quantification [23]. Studies have shown that higher pressure pain sensitivity predicts the efficacy of a transcutaneous electrical nerve stimulation device (TENS) and acupuncture [24,25]. In this study, we used a pain reduction of 50% or more after lidocaine infusion as the criterion for responders. The characteristics of responders, including pain severity, sleep interference, revised fibromyalgia questionnaire (rFIQ) scores, and QST measures, were compared between responders and non-responders.

## Methods

### Study cohort and design

This was an NHS registry study using routinely collected patients' data, which was approved by the R&D Lead of Guy's and St Thomas' Hospital, London. We adhere to RECORD guidelines for reporting studies using observational routinely collected data [26]. This study was a single-centre retrospective analysis of adult patients with fibromyalgia from 2019 to 2023. A clinical diagnosis of fibromyalgia was based on the revised American College of Rheumatology (ACR) 2016 diagnostic criteria [27]. Screening blood tests, including full blood count, and renal and liver function tests, were performed. Patients with autoimmune disease, neurological disorders like painful diabetic neuropathy, allergies to local anaesthetics, severe cardiovascular disease, or impaired liver and renal function were excluded. Individuals could receive conventional treatment, including physiotherapy, transcutaneous electrical nerve stimulation (TENS), a pain management program and medications such as simple analgesics and pregabalin before or after the lidocaine infusion. We have consecutively collected 132 participants who completed pre-treatment questionnaires and QST.

### Questionnaires, screening blood tests, and clinical investigations

The widespread pain index (WPI, 0–19) and symptom severity score (SSS, 0–12) were based on patient-reported symptoms. The severity of pain was rated for the daily worst, least, and average pain using the 11-point numerical rating scale (0 = no pain, 10 = worst possible pain). The daily sleep interference score (DSIS) was measured on a similar rating scale (0 = no interference, 10 = complete interference). The weekly pain and sleep interference scores (WAP and WSIS) were the means of average daily pain (ADP) and DSIS for 7 days. The revised fibromyalgia impact questionnaire (rFIQ) contains 21 items covering 3 domains: functional impairment, overall impact, and symptom severity. All questions were graded on the 11-point numerical rating scale, and the total score of rFIQ is 100. A higher score indicated a greater impact of fibromyalgia on individuals.

### Quantitative sensory testing

The QST was performed on the dorsum of the right foot following a standardised protocol [28]. The following parameters were assessed: cold and heat detection thresholds (CDT and HDT), the ability to detect temperature changes (thermal sensory limen, TSL), cold and heat pain thresholds (CPT and HPT), mechanical detection and pain thresholds (MDT and MPT), mechanical pain sensitivity (MPS), dynamic mechanical allodynia (DMA), wind-up ratio (WUR) and vibration detection. Vibration was recorded as intact or absent. PPT (pressure pain threshold) was not tested consistently, therefore, it was excluded from this study. The results of QST were transformed to Z-scores (patient measurement-mean of age- and gender-matched healthy controls)/SD. Z scores >1.96 or <−1.96, indicating values outside the 95% confidence interval of the baseline values, were considered gain-of-function or loss-of-function, respectively. The combination of sensory gains and losses was analysed using the LOGA classification [29]. Loss of detection for thermal stimuli (CDT or WDT) was coded as L1, for mechanical stimuli (MDT or VDT) as L2, and loss of both as L3. Gain of function for thermal stimuli

(CPT or HPT) was classified as G1, for mechanical stimuli (MPT, MPS, DMA and WUR) as G2, and gain for both as G3. L0 represented no loss for thermal or mechanical detection, and G0 meant no thermal or mechanical allodynia or hyperalgesia. Loss of function of small fibres was indicated by increased CDT and/or WDT, and loss of function of large fibres by increased MDT or loss of vibration detection.

## Lidocaine infusion

Patients were nil-by-mouth for at least 6 hours. A 12-lead ECG was performed, and patients with a QTc > 440 ms in men and > 460 ms in women were excluded. A dose of 5 mg/kg was administered over 1–2 hours using a syringe pump (Injectomat Agilia, Fresenius Kabi). Heart rate, 3-lead ECG and oxygen saturation were monitored continuously throughout the infusion and 30 min after. Blood pressure was taken every 15 minutes.

## Statistical analysis

Lidocaine responders were those who had a 50% or greater reduction in their average weekly pain score within 3–6 weeks after the infusion. A clinically important difference is defined as a 30% pain reduction or a 2-point improvement on the NRS (0–10) [30]. We adopted a stringent 50% pain reduction to identify a group of patients that fell in the category of "very much improvement" and exceeded a 2-point pain reduction. IBM SPSS Statistics 28 was used for statistical analysis. QST parameters were compared with control values corresponding to age, gender, and the dorsum of the foot as a reference site. Using logarithmic transformation of the raw data, Z-scores were calculated as follows: z-score = (value of the subject − mean value of controls)/standard deviation. The 95% confidence interval of healthy controls is between −1.96 and +1.96. Abnormal values were defined as Z-scores outside the 95% confidence interval of healthy controls (< −1.96 = abnormal loss; > 1.96 = abnormal gain). Descriptive statistics were applied to summarise the variables. The continuous variables were reported as mean ± SD (standard deviation), and the discrete variables were expressed as the number of observations and frequency and compared using the Chi-square test. Interval variables, including pain, sleep interference, and rFIQ scores, were expressed as mean ± SD, and compared using Student's *t*-test for 2 groups. All statistical tests were 2-sided, and a *p*-value < 0.05 was considered statistically significant.

## Results

Table 1 presents the demographic and clinical data of the 132 patients in this study, comprising 24 male and 108 female individuals. The diagnostic criteria for fibromyalgia included the total of the widespread pain index (WPI) and systemic symptom severity (SSS) scores. The average WPI and SSS scores were similar for both genders. Female patients were slightly younger and experienced generalised pain earlier than male patients. The female patients had a higher BMI

**Table 1. The demographic and clinical data of fibromyalgia patients in this study.**

|  | All (n = 132) | Male (n = 24) | Female (n = 108) | *p*-value |
|---|---|---|---|---|
| Age (years) | 47.6 ± 11.7 | 51.0 ± 10.4 | 46.9 ± 11.9 | 0.121 |
| BMI (kg/m²) | 30.4 ± 6.9 | 28.0 ± 4.2 | 31.0 ± 7.3 | 0.058 |
| WPI | 14.7 ± 3.9 | 14.2 ± 3.3 | 14.7 ± 4.0 | 0.577 |
| SSS | 9.5 ± 1.9 | 9.0 ± 1.9 | 9.6 ± 1.9 | 0.141 |
| Pain duration (years) | 13.0 ± 8.7 | 13.6 ± 7.6 | 12.9 ± 9.0 | 0.725 |
| Onset of pain (years) | 34.9 ± 11.1 | 37.4 ± 8.7 | 34.4 ± 11.5 | 0.234 |
| Delay of diagnosis (years) | 6.4 ± 5.8 | 6.2 ± 6.0 | 6.5 ± 5.7 | 0.846 |

BMI: body mass index; WPI: widespread pain index; SSS (symptom severity score). Data are expressed as mean ± SD. Student's *t-test* was used to examine the statistical significance between male and female patients. *p < 0.05.

compared to the male patients, but this difference was not statistically significant. The BMI was 28.0±4.2 (range: 19.1–36.3) in the males and 31.0±7.3 (range: 15.5–48.6) in the females ($p=0.058$). On average, it took over 6 years to make the diagnosis of fibromyalgia in this cohort, which represented a marked delay for both genders.

Anti-neuropathic pain medications such as pregabalin, duloxetine and amitriptyline were used by 75% of the patients. Simple analgesics including paracetamol, codeine, dihydrocodeine, ibuprofen and naproxen were taken by 54.2% of the patients. In addition, 36.2% of the patients took opioids, including tramadol, morphine and buprenorphine patches. Anti-neuropathic medication was the most commonly used pain medication in this cohort. 20.8% of male and 4.6% of female patients didn't take any pain medications (Chi-square test $p<0.01$).

The change in daily pain was measured as average daily pain (ADP), worst daily pain (WDP) and least daily pain (LDP). Average weekly pain (AWP) is calculated as the mean of average daily pain (ADP) over 7 consecutive days. Sleep quality was measured using daily and weekly sleep interference scores (DSIS and WSIS). There was no significant statistical difference in pain and sleep scores in the male and female patients (Table 2). The quality of life was measured using a revised fibromyalgia impact questionnaire (rFIQ) that contained 3 categories: functional impairment, impact and symptomatic burden. The female patients reported significantly higher scores in functional impairment. The total scores of rFIQ were 70.9±16.8 in male and 76.9±15.1 in female patients ($p=0.089$), indicating a severely impaired quality of life in both genders.

### Systemic lidocaine infusion was effective, particularly in younger female patients

In general, systemic lidocaine infusion was well tolerated. The most common side effect was dizziness (40%), followed by drowsiness (32%), an increase in diastolic blood pressure (21%) and a tingling sensation around the lips (15%). In instances where side effects manifested, the infusion was temporarily halted, and patients were reassessed at five-minute intervals. The infusion was discontinued in two patients due to persistent symptoms, and their data were excluded from the analysis. Approximately 60% of patients reported pain reduction from 1 to 6 points on the NRS, and the median length of pain relief lasted for 7 weeks, ranging from 1 to 32 weeks (Fig 1A). Approximately a third of patients reported the PGIC as "very much improved" and "much improved", and another one-third of patients reported "no change" (Fig 1B). Five patients reported "slightly worse" or "worse".

Responders to IV lidocaine infusion were defined as those experiencing a pain reduction of 50% or more for at least 3 weeks. Of the 132 patients, 29 were identified as responders, making up 22% of the patient population. One male patient

**Table 2. Scores of pain severity, sleep interference and rFIQ in male and female patients.**

|  | All (n=132) | Male (n=24) | Female (n=108) | p-value |
|---|---|---|---|---|
| ADP (0–10) | 7.6±1.6 | 7.4±1.7 | 7.7±1.6 | 0.457 |
| WDP (0–10) | 8.8±1.3 | 8.5±1.4 | 8.9±1.3 | 0.163 |
| LDP (0–10) | 5.9±2.2 | 5.9±2.5 | 5.9±2.2 | 0.963 |
| AWP (0–10) | 8.3±1.5 | 8.4±1.7 | 8.3±1.5 | 0.883 |
| DSIS (0–10) | 7.5±2.0 | 7.2±2.2 | 7.6±2.0 | 0.371 |
| WSIS (0–10) | 8.1±1.9 | 7.5±2.2 | 8.2±1.8 | 0.138 |
| Function (0–30) | 22.1±6.2 | 19.8±7.7 | 22.6±5.8 | 0.043* |
| Impact (0–20) | 16.7±3.6 | 16.1±3.2 | 16.8±3.7 | 0.375 |
| Symptoms (0–50) | 37.0±7.5 | 34.9±8.8 | 37.4±7.2 | 0.134 |
| rFIQ (0–100) | 75.8±15.5 | 70.9±16.8 | 76.9±15.1 | 0.089 |

ADP: average daily pain; WDP: worst daily pain; LDP: least daily pain; AWP: average weekly pain; DSIS: daily sleep interference score; WSIS: weekly sleep interference score. The revised fibromyalgia impact questionnaire (rFIQ) contains 3 domains: functional impairment (function), impact and symptomatic burden (symptoms). The data are expressed as mean±SD. Student's t-test was used to examine statistical significance between male and female patients.

*$p<0.05$.

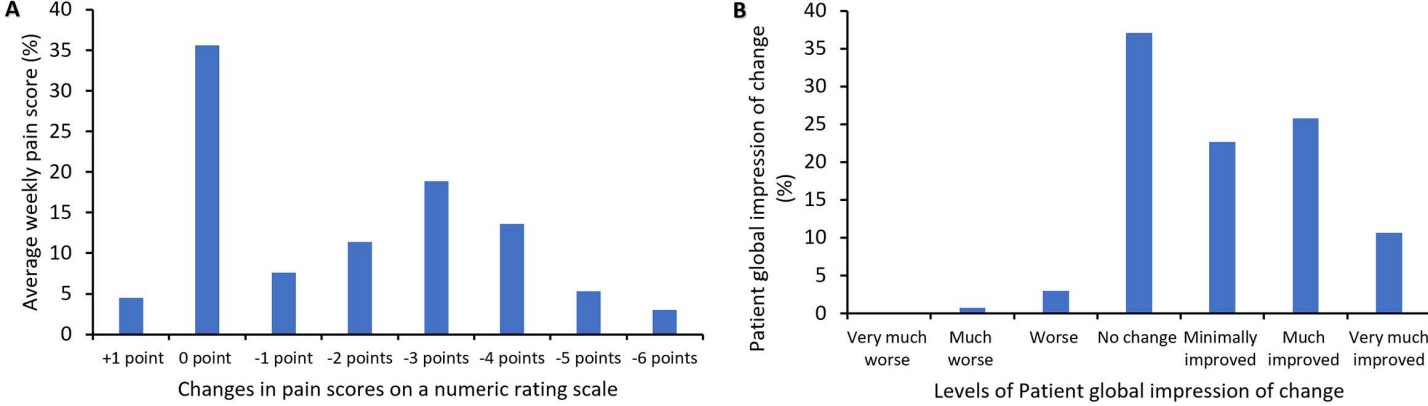

**Fig 1. Changes in average weekly pain score on a numeric rating scale and the patient's global impression of change after intravenous lidocaine infusion.** (A) shows the percentages of patients who reported an increase, no change, or a reduction in average weekly pain score (AWP) at 3 - 6 weeks after infusion. 0 point means no change of AWP, and +1 point represents an increase in pain by 1 point after the infusion. Pain reduction is shown as −1 to −6 points, with −6 being the maximum reduction in this group. (B) shows the percentage of patients reporting various levels of change. Patient global impression of change is a 7-point single-item scale from "very much worse" to "very much improved".

(4.2%) and 28 female patients (25.9%) were responders. A significant gender difference in response to systemic lidocaine was observed ($p = 0.02$, Table 3). Additionally, responders were younger and had a shorter duration of pain compared to non-responders. The mean ages of responders and non-responders were 42.7±11.2 and 49.4±11.4, respectively ($p = 0.003$). The duration of generalised pain was 10.0±6.1 years for responders and 14.1±9.3 years for non-responders ($p = 0.015$). The average time to diagnose fibromyalgia was 4.9±4.6 years for responders and 7.0±6.1 years for non-responders ($p = 0.058$). There was no statistically significant difference in BMI, WPI, or SSS between responders and non-responders.

### Responders tend to report lower pain scores

Responders had a mean AWP of 7.8±1.7, significantly lower than non-responders of 8.5±1.4 ($p = 0.014$). Although ADP, WDP, and LDP were lower in responders, the differences were not statistically significant (Table 4). Responders also showed slightly lower DSIS and WSIS, as well as total rFIQ scores (72.5±16.3 vs. 77.0±15.1, $p = 0.135$).

**Table 3. Characteristics of lidocaine responders in univariate analysis.**

| Factor | Non-responders | Responders | *p*-value |
|---|---|---|---|
| Age (years) | 49.4±11.4 | 42.7±11.2 | 0.003** |
| Male (%) | 95.8 | 4.2 | 0.020* |
| Female (%) | 74.1 | 25.9 | |
| BMI (kg/m²) | 30.9±6.9 | 29.3±6.8 | 0.240 |
| WPI | 14.7±3.9 | 14.6±3.8 | 0.863 |
| SSS | 9.5±1.9 | 9.5±2.0 | 0.964 |
| Age of onset (years) | 35.8±11.0 | 32.7±11.2 | 0.154 |
| Pain duration (years) | 14.1±9.3 | 10.0±6.1 | 0.015* |
| Delay of diagnosis (years) | 7.0±6.1 | 4.9±4.6 | 0.057 |

Responders to lidocaine infusion were defined as patients with a ≥50% drop in the average weekly pain score. BMI: body mass index; WPI: widespread pain index; SSS: symptom severity score. Data are presented as mean±SD. The continuous variables were tested using Student's *t-test,* and the categorical variables were tested with the Chi-square test.

*$p < 0.05$.

**Table 4. Comparison of scores of pain severity, sleep interference and rFIQ between non-responders and responders.**

| | All (n = 132) | Non-responders (n = 103) | Responders (n = 29) | p-value |
|---|---|---|---|---|
| ADP (0–10) | 7.6 ± 1.6 | 7.8 ± 1.5 | 7.3 ± 1.8 | 0.145 |
| WDP (0–10) | 8.8 ± 1.3 | 8.9 ± 1.3 | 8.6 ± 1.4 | 0.174 |
| LDP (0–10) | 5.9 ± 2.2 | 6.0 ± 2.3 | 5.5 ± 1.9 | 0.211 |
| AWP (0–10) | 8.3 ± 1.5 | 8.5 ± 1.4 | 7.8 ± 1.7 | 0.014* |
| DSIS (0–10) | 7.5 ± 2.0 | 7.7 ± 2.1 | 7.3 ± 1.9 | 0.356 |
| WSIS (0–10) | 8.1 ± 1.9 | 8.2 ± 1.9 | 7.7 ± 1.9 | 0.172 |
| Function (0–30) | 22.1 ± 6.2 | 22.5 ± 6.0 | 21.1 ± 3.7 | 0.231 |
| Impact (0–20) | 16.7 ± 3.6 | 17.0 ± 3.0 | 15.9 ± 4.7 | 0.131 |
| Symptoms (0–50) | 37.0 ± 7.5 | 37.4 ± 7.9 | 35.9 ± 6.6 | 0.331 |
| rFIQ (0–100) | 75.8 ± 15.5 | 77.0 ± 15.1 | 72.5 ± 16.3 | 0.136 |

ADP: average daily pain; WDP: worst daily pain; LDP: least daily pain; AWP: average weekly pain; DSIS: daily sleep interference score; WSIS: weekly sleep interference score. The revised fibromyalgia impact questionnaire (rFIQ) contains 3 domains: functional impairment (function), impact and symptomatic burden (symptoms). The data are expressed as mean ± SD. The Student's *t-test* was used to examine statistical significance between responders and non-responders.

* *p* < 0.05.

## Responders and non-responders had similar QST profiles

Fig 2A presents a comparison of mean values of QST parameters between responders and non-responders. Negative deflection appeared in CDT, WDT, TSL, MDT, and to a lesser extent in MPT for both groups. Positive deflection was observed in DMA, WUR, and to a lesser extent in MPS. There was no statistical difference in QST parameters between responders and non-responders. Fig 2B illustrates subtypes of abnormal sensory loss and gain according to the LOGA classification [28]. L0, showing no loss in thermal and mechanical detection, was observed in 42.7% of non-responders and 55.5% of responders. The L3 subtype was present in 26% of non-responders and 22.2% of responders. There was no significant difference in the number of L0, L1, L2, and L3 subtypes between the responders and non-responders (Chi-square test *p* = 0.456). The G0, which exhibited no thermal or mechanical allodynia or hyperalgesia, was the most

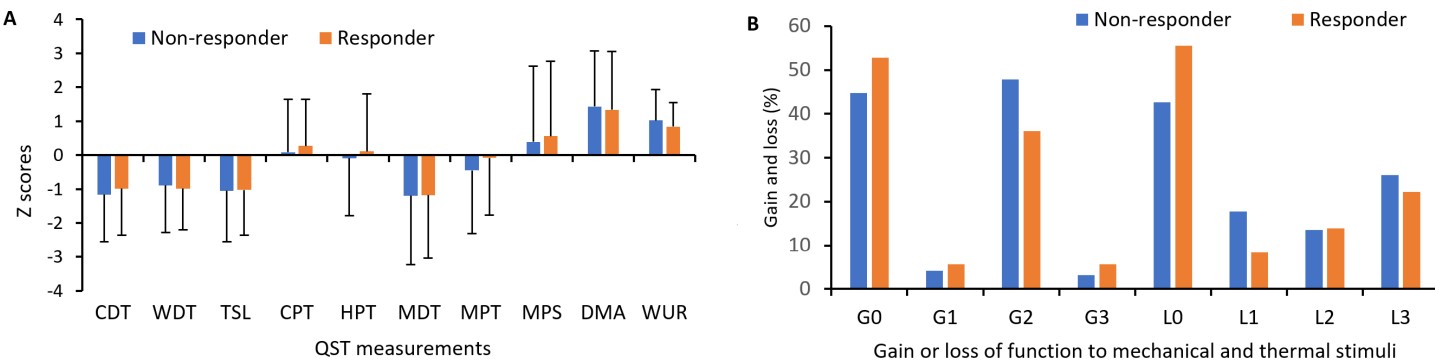

**Fig 2. Comparison of QST modalities and loss/gain phenotypes in responders and non-responders.** (A) shows the following measurement: cold and warm detection thresholds (CDT and WDT), the ability to detect temperature changes (thermal sensory limen, TSL), cold and heat pain thresholds (CPT and HPT), mechanical detection and pain thresholds (MDT and MPT), mechanical pain sensitivity (MPS), dynamic mechanical allodynia (DMA) and wind-up ratio (WUR). The data are expressed as mean ± SD. (B) shows the percentage of patients with loss or gain of function to mechanical and thermal stimuli. Loss of detection for thermal stimuli (CDT or WDT) was coded as L1, for mechanical stimuli (MDT or VDT) as L2, and loss of both as L3. Gain of function for thermal stimuli (CPT or HPT) was coded as G1, for mechanical stimuli (MPT, MPS, DMA and WUR) as G2, and gain for both as G3. L0 represented no loss for thermal or mechanical detection and G0 meant no thermal or mechanical allodynia or hyperalgesia.

common and accounted for 44.8% of non-responders and 52.8% of responders. That was followed by G2 observed in 47.9% and 36.1% of non-responders and responders, respectively (Chi-square test $p = 0.283$). The G1 and G3 were far less common than G0 and G2. G1 and G3 were found in 7.3% of non-responders and 11.2% of responders. The most prevalent QST phenotype among responders was L0G0 (25%), followed by L3G0 (22.2%) and L1G2 (22.2%). In contrast, among non-responders, the predominant phenotype was L3G0 (26.0%), followed by L0G2 (24%) and L0G0 (15.6%).

## Discussion

In this study, we analysed the clinical features and somatosensory profiles of 132 fibromyalgia patients who underwent systemic lidocaine infusion. Our findings were:

- IV lidocaine infusion showed higher effectiveness in female patients compared to male patients.

- Characteristics associated with responders include younger age, shorter pain duration, and lower weekly pain scores.

- No specific QST parameters or loss/gain phenotype are linked to responders.

Systemic lidocaine therapy has provided an alternative approach to refractory neuropathic pain syndromes and fibromyalgia [31]. Lidocaine infusion reduced both evoked and spontaneous neuropathic pain [32,33]. It attenuates peripheral and central sensitisation by blocking sodium channels [17,34–36] and exerts potent anti-inflammatory properties through several mechanisms, including the reduction of circulating inflammatory cytokines [37,38]. There are some variations regarding dose, timing and treatment interval. For example, an RCT pilot study has found that lidocaine infusion at 5 mg/kg, but not 3 mg/kg, was more effective than placebo in relieving neuropathic pain [39]. A higher dose of 7.5 mg/kg had a stronger and longer-lasting effect on pain reduction in patients with fibromyalgia [15]. Systemic lidocaine was administered as a single [11,12], repeated [10,15] or sequential infusions for several days [13,14]. The infusion time varied from 30 minutes to 24 hours. Several RCTs for peripheral and central neuropathic pain have used lidocaine infusion at 5 mg/kg [32,33,40,41]. For this study, we administered a single dose of 5 mg/kg for 1–2 hours and offered repeat infusions to the responders at an interval of 4–6 months. We have found that repeated infusions resulted in a similar degree of pain reduction and functional improvements. However, it is difficult to compare our results directly with those published in previous studies as the protocols of lidocaine infusion and the parameters of assessments were different from one study to another.

There is an inconsistency regarding predictors for responses to lidocaine therapy. Attal and colleagues have found that the severity of mechanical allodynia and the degree of sensory impairment, but not age, pain duration and pain severity, could predict a positive response to lidocaine infusion [36]. A retrospective study suggested that increasing pain intensity and advancing age are predictive factors for the likelihood of pain reduction [42]. We have found that female gender, younger age, short duration of generalised pain and lower weekly pain scores were characteristics of responders. No specific QST parameters or loss/gain phenotype was associated with responders. The discrepancies regarding predictors reflect the complex aetiologies and mechanistic differences of neuropathic pain and fibromyalgia. Our data indicates that early diagnosis and treatment with lidocaine infusion benefits female patients.

Gender influences pain severity and symptomatology in individuals with fibromyalgia. Studies indicate women experience greater pain severity and functional impairment [42–44]. We found no significant difference in pain scores between male and female patients in this cohort, likely due to the small sample size of men (24 out of 132 patients, 18.2%). We have confirmed that female patients reported significantly higher scores in functional impairment. The total score of rFIQ was higher in female patients, but the difference was not statistically significant. In a study involving 352 patients with a similar female/male ratio, Moshrif et al. also reported no difference in rFIQ between female and male patients, which is in agreement with our finding [45]. However, they found that females showed a significantly higher score than males regarding WPI and SSS, but there were no differences in these two indices in our study, which may be due to the differences in age and race. Our patients are significantly older than theirs: females, 47 vs 35, and males, 51 vs 31 years old.

Research indicates that gender may influence the response to treatment. For instance, Arnold and colleagues found that duloxetine is safe and effective for treating fibromyalgia in female patients. However, male patients did not show significant improvement on any efficacy measure [46]. Our study indicates that systemic lidocaine demonstrates greater efficacy in women compared to men. The underlying reasons for these gender differences remain uncertain. Research has demonstrated that lidocaine metabolism does not vary between genders [47]. This disparity may be attributed to potential gender-specific pathophysiological mechanisms in fibromyalgia that influence treatment outcomes.

Evidence suggests that QST is a useful tool for a mechanism-based classification of pain. We have confirmed allodynia, hyperalgesia, and hypoaesthesia to non-nociceptive stimuli in fibromyalgia patients [20–22]. This pattern resembles neuropathic pain but occurs with a different frequency [29]. A recent TwinsUK study found no link between QST modalities and chronic widespread pain, highlighting the complexity of chronic pain syndromes and the limitations of single QST modalities in capturing their diversity [48]. No specific QST modalities or loss/gain phenotypes were identified in association with responders. Further research is necessary to enhance the understanding of the utility of QST in mechanistic pain classification and outcome predictions.

This study has several limitations. The scores for pain severity and quality of life were derived from medical records, and scores for symptoms and questionnaires may have been influenced by patients' medications. Lidocaine infusion was administered alongside routine treatments such as pregabalin, duloxetine, and amitriptyline. QST, a psychophysical assessment, relies on patients' cooperation. Pressure pain threshold and vibration data were excluded due to inconsistent testing. The post-infusion data collection timing varied from 3 to 6 weeks. This may affect the reporting of pain scores and PGIC.

## Conclusion

IV lidocaine infusion, the most common procedure for fibromyalgia in the UK, is particularly effective in women, although the reasons for this gender difference are unclear. Single QST modalities or loss/gain phenotypes were not effective in predicting treatment responses. Our data suggest that systemic lidocaine therapy may be more effective in younger patients. Early diagnosis and treatment with IV lidocaine infusion could particularly benefit younger female patients.

## Supporting information

**S1 Data. Lidocaine QST data set V3.**
(XLSX)

## Acknowledgments

The authors thank Isabel Soares for setting up the lidocaine infusion service and for her dedication to the patients.

## Author contributions

**Conceptualization:** Min Liu.

**Data curation:** Min Liu, Stephany Harris, Anna P. Andreou.

**Formal analysis:** Min Liu.

**Investigation:** Xuenong Bo, Min Liu.

**Methodology:** Xuenong Bo, Min Liu.

**Project administration:** Min Liu.

**Writing – original draft:** Min Liu.

**Writing – review & editing:** Xuenong Bo, Min Liu, Adnan Al-Kaisy, David Pang.

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
