## [Decision Letter · Decision Letter 0]

PONE-D-25-08401Effects and predictors of intravenous lidocaine infusion for patients with fibromyalgiaPLOS ONE

Dear Dr. Bo,

Thank you for submitting your manuscript to PLOS ONE. After careful consideration, we feel that it has merit but does not fully meet PLOS ONE’s publication criteria as it currently stands. Therefore, we invite you to submit a revised version of the manuscript that addresses the points raised during the review process.

Abstract:

- Briefly give the key findings of your study. Include key numeric data (including confidence intervals or *p*  values).

Patients and Methods

-  How were the patients selected (e.g., consecutively, randomly, or selectively)?

- All studies must be reported according to the relevant Equator network guideline: https://www.equator-network.org/

- In your manuscript, you need to state in the Methods section that you have followed relevant Equator guidelines. For example, if you use STROBE: ‘The reporting of this study conforms to STROBE guidelines. (Insert new reference number).

- More details should be provided in terms of the IRB approval.

Discussion

- You concluded that IV lidocaine infusion showed higher effectiveness in female patients compared to

male patients. A comparison of your results and the relevant previous studies showing gender difference in FM should be made. For examples: DOI:10.2147/OARRR.S358255

We look forward to receiving your revised manuscript.

Kind regards,

Wesam Gouda, MD, PhD

Academic Editor

PLOS ONE

**Journal requirements:** 1. When submitting your revision, we need you to address these additional requirements. Please ensure that your manuscript meets PLOS ONE's style requirements, including those for file naming. The PLOS ONE style templates can be found at https://journals.plos.org/plosone/s/file?id=wjVg/PLOSOne_formatting_sample_main_body.pdf and https://journals.plos.org/plosone/s/file?id=ba62/PLOSOne_formatting_sample_title_authors_affiliations.pdf 2. We note that your Data Availability Statement is currently as follows: All relevant data are within the manuscript and its Supporting Information files. Please confirm at this time whether or not your submission contains all raw data required to replicate the results of your study. Authors must share the “minimal data set” for their submission. PLOS defines the minimal data set to consist of the data required to replicate all study findings reported in the article, as well as related metadata and methods (https://journals.plos.org/plosone/s/data-availability#loc-minimal-data-set-definition). For example, authors should submit the following data: - The values behind the means, standard deviations and other measures reported;- The values used to build graphs;- The points extracted from images for analysis. Authors do not need to submit their entire data set if only a portion of the data was used in the reported study. If your submission does not contain these data, please either upload them as Supporting Information files or deposit them to a stable, public repository and provide us with the relevant URLs, DOIs, or accession numbers. For a list of recommended repositories, please see https://journals.plos.org/plosone/s/recommended-repositories. If there are ethical or legal restrictions on sharing a de-identified data set, please explain them in detail (e.g., data contain potentially sensitive information, data are owned by a third-party organization, etc.) and who has imposed them (e.g., an ethics committee). Please also provide contact information for a data access committee, ethics committee, or other institutional body to which data requests may be sent. If data are owned by a third party, please indicate how others may request data access.

Reviewers' comments:

Reviewer's Responses to Questions

**Comments to the Author**

1. Is the manuscript technically sound, and do the data support the conclusions?

Reviewer #1: Yes

Reviewer #2: Yes

Reviewer #3: Yes

2. Has the statistical analysis been performed appropriately and rigorously? 

Reviewer #1: Yes

Reviewer #2: Yes

Reviewer #3: Yes

3. Have the authors made all data underlying the findings in their manuscript fully available?

Reviewer #1: No

Reviewer #2: Yes

Reviewer #3: Yes

4. Is the manuscript presented in an intelligible fashion and written in standard English?

Reviewer #1: Yes

Reviewer #2: Yes

Reviewer #3: Yes

5. Review Comments to the Author

**Reviewer #1: ** The references cited in the manuscript need revision. The author needs to ensure the PLOS ONE journal's guidelines.

For proper formatting, I recommend the authors visit the journal's author guidelines.

The authors identified 50% pain reduction as a reference for responders. I recommend the authors briefly justify the relevance and clinical significance of the 50% pain reduction criterion in the setting of fibromyalgia treatment.

The manuscript did not reflect the characteristics of the control group. While there was mention of controls being matched for age , sex, etc., were there any other control group characteristics that could have had an effect on QST parameters?

It would be interesting if there is a brief description of control group characteristics.

**Reviewer #2:** This manuscript is commendably well-written, exhibiting a clear and coherent structure throughout. The statistical analyses used are appropriate, effectively supporting the study's conclusions. The results are presented with clarity and precision, facilitating easy comprehension of the findings. The discussion section is thoughtfully organized, providing insightful interpretations that are well-grounded in the data. For my specific suggestions, please refer to the PDF attached with this submission.

**Reviewer #3: ** The manuscript “Effects and predictors of intravenous lidocaine infusion for patients with fibromyalgia” by Liu et al. is a retrospective study describing the effect of systematic IV lidocaine infusion on pain relief in 132 patients with fibromyalgia using questionnaires and quantitative sensory testing. The authors identified 22% of patients as responders, with the criteria being patients who had a pain reduction of 50% or greater lasting for at least 3 weeks following an IV lidocaine infusion at a dose of 5 mg/kg; They also found that IV lidocaine infusion was especially effective for younger female patients. This discovery will be incredibly beneficial in designing an effective lidocaine clinical therapy protocol for fibromyalgia patients.

Minor:

1. In the 2nd paragraph of Methods tilted “Questionnaires, screening blood tests,

and clinical investigations”, there are no descriptions about the screening blood tests.

2. Due to the large number of abbreviations within the manuscript, it is advisable to have a table listing all of the abbreviations.

3. The authors set up the responder standard at 50% or greater pain reduction. According to the literature, the responder standards range from 20% to 50% pain reduction. It will be helpful to discuss how the responder standard was chosen to get clinically meaningful data.

6. PLOS authors have the option to publish the peer review history of their article (what does this mean? ). If published, this will include your full peer review and any attached files.

**Do you want your identity to be public for this peer review?** For information about this choice, including consent withdrawal, please see our Privacy Policy .

Reviewer #1: **Yes: ** Vijaya Prasanna Parimi

Reviewer #2: **Yes: ** Clint Vaz

Reviewer #3: No

---

## [Author Response · Author response to Decision Letter 1]

4 Jun 2025

Responses to the editor:

1. Abstract: Briefly give the key findings of your study. Include key numeric data (including confidence intervals or p values).

Key numeric data and p-values have been added to the Abstract.

2. Patients and Methods. How were the patients selected (e.g., consecutively, randomly, or selectively)?

The information on patient selection has been added to the Abstract and the Methods (1st paragraph on page 6).

3. All studies must be reported according to the relevant Equator network guideline: https://www.equator-network.org/ In your manuscript, you need to state in the Methods section that you have followed relevant Equator guidelines. For example, if you use STROBE: ‘The reporting of this study conforms to STROBE guidelines. (Insert new reference number).

Please see the Abstract on page 2 “This study adheres to RECORD guidelines for reporting studies using observational routinely collected data. It was conducted retrospectively and employed data derived from clinical records of patients who received standard care.” and Methods on page 5 “We adhere to RECORD guidelines for reporting studies using observational routinely collected data [26].”

4. More details should be provided in terms of the IRB approval.

Please see Methods on page 5 “This was an NHS registry study using routinely collected patients' data, which was approved by the R&D Lead of Guy's and St Thomas’ Hospital, London.” There isn’t a specific IRB number available.

5. Discussion: You concluded that IV lidocaine infusion showed higher effectiveness in female patients compared to male patients. A comparison of your results and the relevant previous studies showing gender differences in FM should be made. For example: DOI:10.2147/OARRR.S358255

We have added some more discussion. However, it is difficult to compare our results directly with those published in previous studies, as the protocols of lidocaine infusion and the parameters of assessments were different from one study to another.

Response to Reviewers

Reviewer #1: The references cited in the manuscript need revision. The author needs to ensure the PLOS ONE journal's guidelines. For proper formatting, I recommend the authors visit the journal's author guidelines.

We have checked the format of our manuscript following the author’s guidelines closely and made the relevant changes where needed.

The authors identified 50% pain reduction as a reference for responders. I recommend the authors briefly justify the relevance and clinical significance of the 50% pain reduction criterion in the setting of fibromyalgia treatment.

Please see the Methods on page 8. “A clinically important difference is defined as a 30% pain reduction or a 2-point improvement on the NRS (0-10) [30]. We adopted a stringent 50% pain reduction to identify a group of patients that fell in the category of “very much improvement” and exceeded a 2-point pain reduction.”

The manuscript did not reflect the characteristics of the control group. While there was mention of controls being matched for age, sex, etc., were there any other control group characteristics that could have had an effect on QST parameters?

It would be interesting if there is a brief description of control group characteristics

We have compared the two groups: responders and non-responders, based on the responses to IV lidocaine infusion. QST parameters and phenotypes were similar in responders and non-responders. Responders were mostly females and significantly younger than non-responders. The raw data of QST was transformed to Z scores. Please see the Methods on page 7. “The results of QST were transformed to Z-scores (patient measurement-mean of age- and gender-matched healthy controls )/SD”

Minor:

1. In the 2nd paragraph of Methods titled “Questionnaires, screening blood tests,

and clinical investigations”, there are no descriptions about the screening blood tests.

Please see Methods on page 6. Screening blood tests, including full blood count, and renal and liver function tests, were performed.

2. Due to the large number of abbreviations within the manuscript, it is advisable to have a table listing all of the abbreviations.

We have added a list of abbreviations based on the suggestions. Please see the abbreviations on page 3.

3. The authors set up the responder standard at 50% or greater pain reduction. According to the literature, the responder standards range from 20% to 50% pain reduction. It will be helpful to discuss how the responder standard was chosen to get clinically meaningful data.

Please see the Methods on page 8. “A clinically important difference is defined as a 30% pain reduction or a 2-point improvement on the NRS (0-10) [30]. We adopted a stringent 50% pain reduction to identify a group of patients that fell in the category of “very much improvement” and exceeded a 2-point pain reduction.”

Reviewer #2:

We have made changes based on the comments made in the PDF file.

---

## [Decision Letter · Decision Letter 1]

Effects and predictors of intravenous lidocaine infusion for patients with fibromyalgia

PONE-D-25-08401R1

Dear Dr. Bo,

We’re pleased to inform you that your manuscript has been judged scientifically suitable for publication and will be formally accepted for publication once it meets all outstanding technical requirements.

Kind regards,

Wesam Gouda, MD,PhD

Academic Editor

PLOS ONE

Reviewers' comments:

Reviewer's Responses to Questions

**Comments to the Author**

1. If the authors have adequately addressed your comments raised in a previous round of review and you feel that this manuscript is now acceptable for publication, you may indicate that here to bypass the “Comments to the Author” section, enter your conflict of interest statement in the “Confidential to Editor” section, and submit your "Accept" recommendation.

Reviewer #1: All comments have been addressed

Reviewer #2: All comments have been addressed

Reviewer #3: All comments have been addressed

2. Is the manuscript technically sound, and do the data support the conclusions?

Reviewer #1: Yes

Reviewer #2: Yes

Reviewer #3: Yes

3. Has the statistical analysis been performed appropriately and rigorously? 

Reviewer #1: Yes

Reviewer #2: Yes

Reviewer #3: Yes

4. Have the authors made all data underlying the findings in their manuscript fully available?

Reviewer #1: Yes

Reviewer #2: Yes

Reviewer #3: Yes

5. Is the manuscript presented in an intelligible fashion and written in standard English?

Reviewer #1: Yes

Reviewer #2: Yes

Reviewer #3: Yes

6. Review Comments to the Author

Reviewer #1: ---------------------------------------------------- ----------------------------nil-------------------------------------------------

Reviewer #2: The revised manuscript is now well written, clear, and methodologically sound. The authors have adequately addressed prior concerns, and the presentation of results is coherent and supported by appropriate references. The study offers meaningful insights and is suitable for publication in its current form. I have no concerns regarding dual publication, research ethics, or publication integrity.

Reviewer #3: The authors already gave clear answers to all my questions including adding an abbreviation information.... I have no more questions.

7. PLOS authors have the option to publish the peer review history of their article (what does this mean? ). If published, this will include your full peer review and any attached files.

**Do you want your identity to be public for this peer review?** For information about this choice, including consent withdrawal, please see our Privacy Policy .

Reviewer #1: **Yes: ** Vijaya Prasanna Parimi ----------------------------------------

Reviewer #2: **Yes: ** Clint Vaz

Reviewer #3: **Yes: ** Gerald Z Zhuang

---

## [Editor Report · Acceptance letter]

PONE-D-25-08401R1

PLOS ONE

Dear Dr. Bo,

I'm pleased to inform you that your manuscript has been deemed suitable for publication in PLOS ONE. Congratulations! Your manuscript is now being handed over to our production team.

Kind regards,

on behalf of

Dr. Wesam Gouda

Academic Editor

PLOS ONE